# Comparison of Anti-Obesity Effects of Ginger Extract Alone and Mixed with Long Pepper Extract

**DOI:** 10.3390/biomedicines13092077

**Published:** 2025-08-26

**Authors:** Gunju Song, Hyein Han, Heegu Jin, Jongwon Kim, Hyeongmin Kim, Yi-Seul Seo, Heewon Song, Boo-Yong Lee

**Affiliations:** 1Department of Food Science and Biotechnology, College of Life Science, CHA University, Seongnam 13488, Republic of Korea; juhun022188@naver.com (G.S.); hyeinoo@naver.com (H.H.); heegu94@hanmail.net (H.J.); nanananakim@gmail.com (J.K.); 2Ju Yeong NS Co., Ltd., Seoul 05854, Republic of Korea; hmkim@juyeongns.com (H.K.); sysns@juyeongns.com (Y.-S.S.); hwsong@juyeongns.com (H.S.)

**Keywords:** browning, lipid accumulation, thermogenesis, ginger extract, long pepper extract, metabolic syndrome

## Abstract

**Background/Objectives:** Obesity is a chronic metabolic disorder characterized by the excessive expansion of adipose tissue and impaired energy homeostasis. Natural products, such as plant extracts, are gaining attention as potential anti-obesity agents. This study aimed to evaluate and compare the anti-obesity effects of ginger (*Zingiber officinale* Roscoe) extract alone and as a mixture with long pepper (*Piper longum* L.) extract in a mouse model of high-fat diet-induced obesity. **Methods:** Male ICR mice were fed a high-fat diet to induce obesity and were orally administered ginger extract (60 mg/kg/day) or a 1:1 mixture of ginger and long pepper extracts (30 mg/kg/day each) for 8 weeks. Body weight, fat mass, glucose tolerance, and serum lipid levels were measured. **Results:** Ginger extract alone significantly reduced body weight gain and visceral and subcutaneous fat accumulation and improved glucose homeostasis and serum lipid profiles compared to the high-fat diet group. These effects were more pronounced than those observed with the mixture group. Ginger extract upregulated lipolytic markers via activation of the protein kinase A (PKA) signaling pathway and increased expression of uncoupling protein 1 (UCP1), indicating browning of white adipose tissue. **Conclusions:** Ginger extract alone exhibited significant anti-obesity effects compared to the mixture with long pepper extract. These findings suggest that ginger extract may serve as a promising natural agent for the prevention and management of obesity-related metabolic dysfunction.

## 1. Introduction

Obesity is a complex and chronic metabolic disorder characterized by the excessive accumulation of adipose tissue and the disruption of normal glucose homeostasis [1,2,3]. This abnormal fat accumulation not only indicates energy imbalance but also leads to various metabolic dysfunctions [4]. As fat mass increases, particularly in visceral fat, it induces a state of low-grade, chronic inflammation that interferes with insulin action, exacerbating insulin resistance and impairing glucose homeostasis [5]. Over time, this metabolic dysregulation significantly increases the risk of developing various conditions associated with metabolic syndrome, including hyperlipidemia, hypertension, impaired glucose tolerance, and non-alcoholic fatty liver disease (NAFLD) [6]. These comorbidities are major risk factors for cardiovascular disease and type 2 diabetes [7].

White adipose tissue (WAT), traditionally considered as merely an energy reservoir, is now recognized as a highly dynamic endocrine organ that secretes a variety of adipokines and inflammatory cytokines, which regulate lipid and glucose metabolism [5,8]. The expansion of WAT occurs through two primary mechanisms: hypertrophy, characterized by an increase in the size of adipocytes, and hyperplasia, defined as an increase in the number of adipocytes [9,10,11]. Hyperplasia is regulated at the transcriptional level by key adipogenic factors such as CCAAT/enhancer-binding protein alpha (C/EBPα), peroxisome proliferator-activated receptor gamma (PPARγ), and fatty acid-binding protein 4 (FABP4), which coordinate the differentiation of preadipocytes into fully differentiated adipocytes enriched in intracellular lipids [12,13,14]. During hypertrophy, mature adipocytes accumulate lipids, a process in which lipogenic enzymes such as lysophosphatidic acid acyltransferase theta (LPAATθ), lipin 1, and diacylglycerol acyltransferase 1 (DGAT1) play a critical role [15,16]. These enzymes catalyze the sequential reactions involved in triglyceride biosynthesis, facilitating lipid storage in adipose tissue [17]. Lipolysis is the catabolic breakdown of triglycerides into free fatty acids (FFAs) and glycerol, thereby supporting energy metabolism [18,19,20]. This process is primarily initiated by protein kinase A (PKA), which activates a lipase cascade involving adipose triglyceride lipase (ATGL), phosphorylated hormone-sensitive lipase (p-HSL), and monoacylglycerol lipase (MGL) [21]. The liberated FFAs can be oxidized within adipocytes or transported to peripheral tissues such as the liver, heart, and skeletal muscle for energy utilization [22]. In contrast to energy storage, brown adipose tissue (BAT) dissipates energy as heat via non-shivering thermogenesis mediated by uncoupling protein 1 (UCP1) [23,24]. Under specific stimuli, white adipocytes can undergo a phenotypic transition known as “browning,” which is characterized by the upregulation of thermogenic genes including peroxisome proliferator-activated receptor alpha (PPARα), PPARγ coactivator 1-alpha (PGC1α), PR domain-containing 16 (PRDM16), and UCP1, offering a potential mechanism to increase energy expenditure and combat obesity [25,26].

Ginger (*Zingiber officinale* Roscoe) extract (GE) has consistently demonstrated anti-obesity effects and is regarded as a promising candidate for managing diet-induced metabolic disorders [27,28,29,30]. Based on previous studies, GE was administered at 60 mg/kg/day to evaluate its metabolic efficacy. Long pepper (*Piper longum* L.) extract (LPE), which contains the bioactive compound piperine, has also shown potential to modulate lipid metabolism and enhance metabolic function [31,32]. Although both GE and LPE possess metabolic regulatory potential, their combined effects have not been systematically evaluated, and therefore their combined effects remain unclear.

To address this gap, we investigated the anti-obesity effects of GE alone and in combination with LPE in an HFD-induced obesity mouse model. We specifically co-administered LPE with GE at a 1:1 ratio, with the LPE dose (30 mg/kg/day) determined based on the acceptable daily intake (ADI) of piperine for humans, as defined by the European Food Safety Authority (EFSA), and was converted to an equivalent dose for mice [33]. This dose was within the safe intake established by the revised no-observed-adverse-effect level (NOAEL). By assessing body weight, fat mass, glucose and lipid profiles, adipokine secretion, histological changes, and key proteins related to adipose tissue function, this study aimed to clarify whether co-administration provides synergistic benefits compared to GE alone in ameliorating HFD-induced obesity.

## 2. Materials and Methods

### 2.1. Preparation and Standardization of Ginger and Long Pepper Extracts

The GE was used as Ginginoll, which was manufactured by AKAY Natural Ingredients Private Limited (Kerala, India) and provided by Ju Yeong NS Co., Ltd. (Seoul, Republic of Korea). To confirm the synergistic effect with the GE, LPE used in the mixture (GE+LPE) was manufactured by Ju Yeong NS Co., Ltd. and is standardized to Piperine ≤ 2%. For this experiment, it was administered after being mixed with the ginger extract in a 1:1 ratio based on weight for use.

GE was prepared from the dried rhizomes of *Zingiber officinale* Roscoe, and LPE was derived from the dried fruits of *Piper longum* L. GE was prepared via supercritical CO_2_ extraction with operating conditions in the range of 25–30 °C, 8–10 bar, 2–4 h, followed by blending with arabic gum and maltodextrin and spray-drying. More detailed extraction parameters are proprietary information(know-how) about the manufacturer. LPE was prepared by 65% ethanol extraction of dried long pepper fruits, followed by concentration, blending with arabic gum, spray-drying, and final standardization. The nutritional composition and marker compounds of both extracts were analyzed to assess functional characteristics (Table 1). Total polyphenol content (TPC) was measured using the Folin–Denis method, and the results were expressed as mg gallic acid equivalents (GAE) per gram of extract.

Marker compounds were quantified by high-performance liquid chromatography (HPLC) equipped with a photodiode array detector (PDA). Analyses were performed using a Waters Arc HPLC series equipped with a C18 column (4.6 × 250 mm, 5 µm; Luna C18(2)). The mobile phase was acetonitrile and distilled water with 1% acetic acid (65:35, isocratic) at 1.0 mL/min. The detection wavelength was set at 280 nm, the column was maintained at 35 °C, and the injection volume was 10 µL. Under these conditions, the retention times were 5.036 min for 6-gingerol and 5.694 min for piperine. The concentrations of 6-gingerol in GE and piperine in LPE were 132 mg/g and 20 mg/g, respectively. A representative HPLC chromatogram of each extract is shown in the Appendix A.

### 2.2. Animals and Treatments

The animal studies were approved by the Institutional Animal Care and Use Committee of CHA University (approval number: IACUC230174). Five-week-old male ICR mice, an outbred strain, were purchased from Raon Bio (Yongin, Republic of Korea) and housed at 20 ± 3 °C in a room maintained under a 12 h light/12 h dark cycle. After a one-week period of adaptation, the mice were randomly assigned in a blinded manner into four groups (*n* = 14 per group): a chow diet (Ctrl) group, a high-fat diet (HFD) group, a group fed HFD supplemented with oral GE at 60 mg/kg/day (GE), or a group fed HFD supplemented with GE and LPE co-administered at 30 mg/kg/day each (GE+LPE). Mice were fed with HFD for 8 weeks. GE or GE+LPE administration was initiated simultaneously with HFD feeding and continued for 8 weeks to assess both preventive and therapeutic effects. The HFD contained 60 kcal% as fat (D12492, Research Diets, New Brunswick, NJ, USA), while the chow diet contained 10 kcal% as fat (D12450B, Research Diets, New Brunswick, NJ, USA). Mice were fed both diets for 8 weeks, with weekly measurements of body mass and dietary intake. At the end of the experiment, the mice were euthanized by exposure to carbon dioxide (CO_2_) until respiratory arrest after fasting for 12 h, and then blood and tissue samples were collected.

### 2.3. Fasting Blood Glucose Measurement

The fasting glucose concentrations were measured weekly from the tail vein after 12 h of fasting using a blood glucose test meter (Accu-Chek, Roche Diagnostics, Basel, Switzerland). We selected weekly measurements to monitor the development of diet-induced hyperglycemia while minimizing stress from repeated bleeding. This approach is commonly employed in dietary intervention studies [34,35,36,37,38,39,40] and allows for effective monitoring of treatment-related changes.

### 2.4. Oral Glucose Tolerance Test and Insulin Tolerance Test

The oral glucose tolerance test (OGTT) was conducted after a 12-h fast. D-glucose (1.5 g/kg body weight) was orally administered, and blood glucose concentrations were measured at 0 (baseline), 30, 60, 90, and 120 min after administration using a glucose meter (Accu-Chek, Roche Diagnostics, Basel, Switzerland). Additionally, an Insulin Tolerance Test (ITT) was performed following the same 12-h fasting period. Mice received an intraperitoneal injection of insulin (1 U/kg body weight), and blood glucose concentrations were measured at 0 (baseline), 30, 60, 90, 120, and 150 min following the injection.

### 2.5. Histological Analysis

Samples of subcutaneous (sWAT) and visceral (vWAT) WAT samples were fixed in 4% paraformaldehyde and embedded in paraffin. Several sections were then prepared and stained with hematoxylin and eosin (H&E) for histological assessment. Photomicrographs were obtained using a Nikon E600 microscope (Nikon, Tokyo, Japan).

### 2.6. Rectal Temperature Measurement

The rectal temperatures of the mice were measured weekly using a Testo 925 Type Thermometer (Testo, Lenzkirch, Germany).

### 2.7. Immunofluorescence Staining

Samples of sWAT and vWAT sections were deparaffinized and then incubated with anti-PKA or anti-UCP1 antibodies. Subsequently, secondary anti-mouse fluorescein isothiocyanate (FITC)-conjugated and anti-rabbit Alexa Fluor™ 594-conjugated antibodies were then applied. DAPI (Thermo Fisher Scientific, Waltham, MA, USA) was used to stain the cell nuclei, and the sections were mounted with ProLong Gold Antifade reagent (Thermo Fisher Scientific). Fluorescent images were captured using a Zeiss confocal laser scanning microscope (LSM880; Carl Zeiss, Oberkochen, Germany) along with Zen 3.10 software (Carl Zeiss).

### 2.8. Biochemical Analysis

Blood samples were obtained through cardiac puncture under terminal anesthesia and centrifuged at 3000× *g* for 20 min at 4 °C to separate the serum. The serum concentrations of insulin, leptin, and resistin were measured using a Mouse Adipokine Magnetic Bead Panel (Merck Millipore, Burlington, MA, USA). The serum concentrations of the total GLP-1 were measured using a Metabolic Hormone Panel V3 (Merck Millipore, Burlington, MA, USA). Additionally, the serum concentrations of triglycerides (TG), total cholesterol, low-density lipoprotein (LDL)-cholesterol, and high-density lipoprotein (HDL)-cholesterol, as well as the activities of aspartate aminotransferase (AST) and alanine aminotransferase (ALT), were measured using colorimetric assay kits from Roche. The sample size for each assay varies depending on sample availability and quality. Therefore, the exact sample size (*n*) for each measurement is explicitly stated in the corresponding figure legend.

### 2.9. Oil Red O Staining

Cryostat sections of liver tissue (5 μm) were stained with a 0.1% (*m*/*v*) oil red o (oro) solution to visualize lipid accumulation in the liver. After staining at room temperature, the sections were rinsed and examined to evaluate the presence and distribution of lipid droplets.

### 2.10. Western Blot Analysis

Tissues were washed twice with phosphate-buffered saline (PBS) and then lysed using a lysis buffer that included 1 mM phenylmethylsulfonyl fluoride, 1 mM ethylenediaminetetraacetic acid, 1 μM pepstatin A, 1 μM leupeptin, and 0.1 μM aprotinin (iNtRON Biotechnology, Seoul, Republic of Korea), along with phosphatase and protease inhibitors. The samples were allowed to stand on ice for 1 h to facilitate lysis. Following homogenization, the samples were centrifuged at 13,000× *g* for 20 min at 4 °C. The protein content in the supernatant was determined, and the lysate protein concentrations were quantified using a protein assay kit (Bio-Rad, Hercules, CA, USA). Lysates containing equal amounts of protein were separated using sodium dodecyl sulfate-polyacrylamide gel electrophoresis (SDS-PAGE), and the proteins were electrotransferred to membranes. The membranes were then blocked with 5% skim milk for 1 h, washed with Tris-buffered saline containing Tween 20 (TBST), and incubated with primary antibodies overnight at 4 °C. Subsequently, the membranes were exposed to horseradish peroxidase-conjugated secondary antibodies. Antibodies targeting C/EBPα, PPARγ, FABP4, sterol regulatory element-binding protein 1 (SREBP1), LPAATθ, lipin 1, DGAT1, phosphorylated PKA (p-PKA, Ser 114), α-tubulin, and PGC1α were purchased from Santa Cruz Biotechnology (Dallas, TX, USA). Antibodies targeting ATGL, phosphorylated HSL (p-HSL, Ser 563), and Fatty Acid Synthase (FAS) were purchased from Cell Signaling Technology (Danvers, MA, USA). Antibodies targeting MGL, PPARα, PRDM16, and UCP1 were purchased from Abcam (Cambridge, UK).

### 2.11. Statistical Analysis

Data are expressed as mean ± SEM. Statistical comparisons were made using one-way ANOVA followed by Tukey’s post-hoc test (IBM SPSS Statistics Version 20.0, Armonk, NY, USA). *p* < 0.05 was regarded as indicating statistical significance.

## 3. Results

### 3.1. GE More Effectively Suppresses HFD-Induced Obesity than GE+LPE

To evaluate the anti-obesity efficacy of GE and its combination with LPE (GE+LPE), mice were fed an HFD for 8 weeks and administered either GE alone or the GE+LPE mixture. At the end of the treatment period, the body mass of mice in the HFD group was substantially higher than in the Ctrl group (Figure 1A,B). Both treatment groups showed reduced body mass compared to the HFD group. However, the GE group exhibited a significantly greater reduction in body weight gain compared to the GE+LPE group. This pattern was consistent in both body weight and adipose tissue measurements (Figure 1C,D). The masses of vWAT, sWAT, and perirenal WAT were significantly lower in the GE group compared to the GE+LPE group. In contrast, no significant differences were observed in the masses of organs such as the kidneys, lungs, and spleens across the four groups (Figure 1E). Moreover, no significant differences in food or water intake were observed among the HFD-fed groups (Figure 1F), suggesting that the observed differences in body weight were not due to reduced energy intake.

### 3.2. GE More Effectively Improves Glucose Homeostasis and Metabolic Hormone Regulation than GE+LPE

To further evaluate the metabolic effects of GE and GE+LPE, we next assessed glucose intolerance and insulin resistance. Over the 8 weeks of the experiment, the fasting blood glucose concentrations of the HFD group increased gradually, while both treatment groups showed a significant decrease in fasting blood glucose concentrations compared to the HFD group. Notably, the GE group showed a more pronounced reduction than the GE+LPE group (Figure 2A). To assess glucose homeostasis, OGTT and ITT were conducted after the 8-week treatment period. As shown in Figure 2B, both the GE and GE+LPE groups displayed significantly lower fasting glucose concentrations and reduced area under the curve (AUC) during the OGTT compared to the HFD group, with the GE group demonstrating a more pronounced improvement in glucose regulation. Similarly, the results of the ITT revealed that blood glucose concentrations declined more rapidly in both treatment groups than in the HFD group (Figure 2C). Consistent with these results, serum insulin concentrations were significantly lower in both treatment groups compared to HFD controls, particularly in the GE group (Figure 2D). Additionally, serum concentrations of glucagon-like peptide-1 (GLP-1), an incretin hormone that promotes insulin secretion [41], were significantly elevated in the GE group (Figure 2E).

In addition, GE treatment significantly improved the dysregulated lipid profile induced by HFD, including reductions in triglycerides (TG), total cholesterol, and LDL cholesterol, as well as higher concentrations of HDL cholesterol (Figure 3A–D). Furthermore, the administration of GE significantly reduced the concentrations of leptin and resistin in serum (Figure 3E,F), which are adipokines closely associated with obesity-related metabolic dysfunction. These findings collectively suggest that GE exerts a more pronounced impact on improving glucose homeostasis, lipid profiles, and metabolic hormones compared to GE+LPE in HFD-induced obese mice.

### 3.3. GE More Effectively Inhibits Adipogenesis and Lipogenesis in WAT than GE+LPE

H&E staining demonstrated that adipocytes in both sWAT and vWAT of the HFD group were markedly larger than those in the Ctrl group (Figure 4A). The quantification confirmed that both the GE and GE+LPE groups significantly reduced adipocyte size, with a more pronounced effect observed in the GE group (Figure 4B,C). To investigate the underlying mechanisms, the expression of key regulators of adipogenesis (C/EBPα, PPARγ, and FABP4) and lipogenesis (LPAATθ, lipin 1, and DGAT1) was assessed in sWAT and vWAT by western blot analysis. Expression levels of these proteins were increased in the HFD group compared to the Ctrl group. However, the GE group showed a significant suppression of these adipogenic and lipogenic factors than the GE+LPE group (Figure 4D,E). These results indicate that GE more effectively inhibits adipocyte differentiation and lipid accumulation in WAT compared to GE+LPE, thereby contributing to its anti-obesity effects in HFD-induced obesity.

### 3.4. GE More Effectively Promotes Lipolysis and Browning in WAT than GE+LPE

To evaluate the thermogenic effects, weekly measurements of rectal temperature were performed. The GE group showed significantly higher rectal temperatures than both the HFD and GE+LPE groups, which may reflect increased thermogenic activity (Figure 5A). Consistent with this finding, immunofluorescence staining showed that the intensity of PKA and UCP1 signals was markedly increased in WAT from the GE group compared to the HFD and GE+LPE groups (Figure 5B). To further investigate the molecular mechanisms underlying these effects, we assessed the expression of lipolytic enzymes, including ATGL, p-HSL, and MGL. Both treatment groups exhibited upregulated expressions of these proteins compared to the HFD group, with the GE group displaying a more pronounced increase in lipolytic activity. Additionally, the expression of thermogenic genes such as PPARα, PGC1α, PRDM16, and UCP1 was elevated in both treatment groups, and GE induced a more significant upregulation than GE+LPE (Figure 5C,D). Taken together, these findings suggest that GE promotes WAT browning and thermogenesis more effectively than GE+LPE, likely due to enhanced lipolysis and activation of key thermogenic pathways.

### 3.5. GE More Effectively Ameliorates Hepatic Steatosis than GE+LPE

To determine whether GE or GE+LPE could improve obesity-associated hepatic steatosis, we examined liver morphology and lipid accumulation. HFD feeding induced significant hepatic steatosis, as evidenced by pale and enlarged livers. In contrast, livers from the GE group exhibited a darker, healthier appearance compared to the Ctrl group (Figure 6A). Moreover, serum concentrations of biochemical factors of liver dysfunction, alanine aminotransferase (ALT) and aspartate aminotransferase (AST), were significantly lower in the GE group than in both the HFD and GE+LPE groups (Figure 6B,C). In addition, liver mass was significantly reduced in the GE group compared to both the HFD and GE+LPE groups (Figure 6D). Consistent with these findings, Oil Red O staining showed extensive lipid accumulation in the HFD group, whereas it was more substantially reduced in the GE group than in the GE+LPE group (Figure 6E).

We next examined hepatic expression of key lipogenic proteins, including SREBP1, FAS, LPAATθ, lipin1, and DGAT1. Although both groups showed reduced expression of lipogenic proteins (Figure 6F), GE induced a significant suppression compared to GE+LPE. This suggests that GE more effectively inhibits hepatic lipid synthesis and steatosis under HFD conditions.

## 4. Discussion

The study demonstrated that GE administration significantly ameliorates obesity in HFD-fed mice compared to its combination with LPE. While both GE and GE+LPE treatments led to reductions in body weight gain, adipose tissue mass, and lipid accumulation, the GE group consistently exhibited more significant effects across key metabolic indicators. These results suggest that the anti-obesity potential of GE may not be synergistically enhanced by co-administration with LPE under the tested dose and ratio.

Beyond weight and lipid regulation, improvements in glucose tolerance and insulin sensitivity were observed in both treatment groups, and these improvements were more significant in the GE group. The observed effects may primarily result from secondary metabolic improvements resulting from the reduction of excessive adipose tissue. Consistently, OGTT and ITT results demonstrated more rapid and sustained reductions in blood glucose concentrations in the GE group compared to the GE+LPE group, indicating enhanced glucose utilization and insulin sensitivity. This was accompanied by an increase in circulating insulin concentrations. In addition, GLP-1 concentrations were also elevated in the GE group, which may have contributed to the observed improvements in glucose homeostasis by enhancing insulin secretion [42,43]. Moreover, the GE group exhibited improved lipid profiles and reduced circulating concentrations of leptin and resistin, supporting its metabolic efficacy compared to GE+LPE [44].

At the tissue level, GE treatment more effectively suppressed both adipogenesis and lipogenesis in WAT. This was reflected not only in the reduction of adipocyte size but also in the decreased expression of transcription factors involved in adipocyte differentiation (C/EBPα, PPARγ, and FABP4) and enzymes regulating lipid synthesis (LPAATθ, lipin 1, and DGAT1). These changes were also accompanied by improvements in hepatic steatosis, possibly due to improved lipid metabolism. Alongside these effects, GE enhanced lipolytic activity, as shown by increased levels of ATGL, p-HSL, and MGL. It also promoted thermogenic reprogramming, evidenced by elevated expression of UCP1 and its upstream regulators PPARα, PGC1α, and PRDM16 [45]. Elevated rectal temperatures and enhanced UCP1 immunoreactivity further substantiated the interpretation that GE promotes energy expenditure through thermogenic activation. These coordinated effects suggest that GE facilitates a functional shift in WAT from lipid storage toward energy dissipation.

The metabolic benefits observed with GE treatment in our study are largely consistent with previous reports on ginger in metabolic disease models. For example, ginger treatment has been shown to reduce body weight gain and WAT mass [28,46,47], along with improving lipid and glucose profiles [48,49]. Furthermore, ginger has been reported to promote markers of browning in HFD models, which aligns with our observation of increased UCP1 and PGC-1α expression in GE-treated mice [50,51]. However, despite reports that piperine can enhance bioavailability and modulate lipid metabolism under specific experimental conditions [52,53], the co-administration of LPE at the 1:1 ratio did not provide additional metabolic benefits beyond GE alone. The ginger extract used in this study was obtained via supercritical CO_2_ extraction, which enriches non-polar bioactive components. The higher abundance of these constituents in supercritical extracts may partly explain the significant metabolic efficacy observed with GE alone.

Several limitations should be acknowledged. First, only a single dose and ratio of GE+LPE were tested. Although piperine is known to enhance bioavailability and modulate lipid metabolism, co-administration with GE did not produce synergistic effects, possibly due to the conservative LPE dose based on the EFSA-established ADI. Different dose combinations or longer treatment durations may yield different outcomes. Second, the sample size for some analyses was reduced due to sample availability or technical exclusions, as described in Section 2. Future studies with a larger sample size and systematic study designs are warranted.

Overall, these findings indicate that GE alone exerts more significant anti-obesity effects under HFD conditions, while LPE co-administration did not enhance its metabolic benefits at the tested dose and ratio. The current results highlight the clinical potential of GE as an individual, naturally derived therapeutic agent for obesity-related metabolic disorders, suggesting a simplified and practical therapeutic approach.

## 5. Conclusions

The present study demonstrated that GE alone exerted more pronounced anti-obesity effects than the GE+LPE mixture in HFD-induced obese mice. GE significantly attenuated body weight gain, reduced visceral and subcutaneous fat accumulation, improved glucose tolerance and serum lipid profiles, and promoted energy metabolism by enhancing the expression of browning-related proteins. These findings indicate that GE may represent a promising natural candidate for the prevention of diet-induced metabolic dysfunction. Although the 1:1 ratio of GE and LPE was selected based on prior evidence, synergistic effects were not observed. Future studies are encouraged to investigate alternative dosing strategies to clarify the potential interactions with piperine.

## Figures and Tables

**Figure 1 biomedicines-13-02077-f001:**
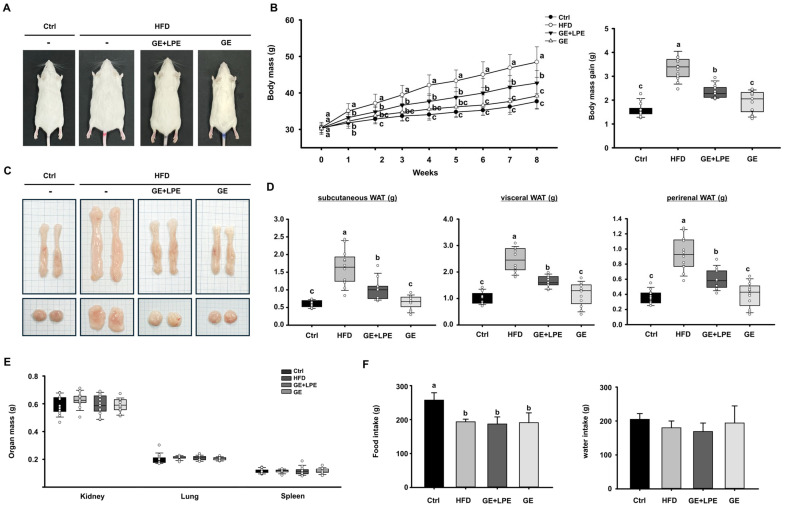
Effects of GE and GE+LPE on body weight, fat mass, and energy intake in HFD-fed mice. (**A**) Representative images of the mice from each group at the end of the 8-week treatment. (**B**) Body mass was measured regularly during the 8-week treatment, and body mass gain over the 8 weeks (mean ± SEM; *n* = 14). (**C**) Representative images of dissected sWAT (**top**) and vWAT (**bottom**). (**D**) Masses of the sWAT, vWAT, and perirenal WAT (mean ± SEM; *n* = 13). (**E**) Masses of other organs, including kidneys, lungs, and spleen (mean ± SEM; *n* = 13). (**F**) Food and water intake were recorded during the 8-week treatment. Values with different letters are significantly different: *p* < 0.05 (a > b > c).

**Figure 2 biomedicines-13-02077-f002:**
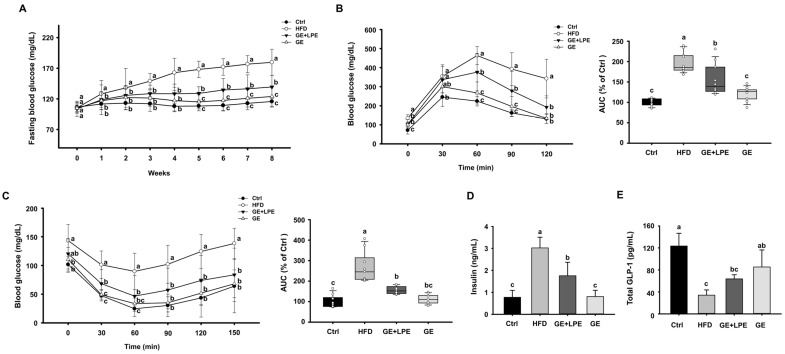
Effects of GE and GE+LPE on the glucose intolerance and insulin resistance of HFD-fed mice. (**A**) Fasting blood glucose concentrations during the 8-week treatment (mean ± SEM; *n* = 12). (**B**) Results of oral glucose tolerance test (OGTT) after 8-week treatment, and corresponding areas under the curves (AUC) (mean ± SEM; *n* = 13). (**C**) Results of insulin tolerance testing (ITT) after 8-week treatment and corresponding AUC (mean ± SEM; *n* = 10). (**D**) Serum insulin concentrations (mean ± SEM; *n* = 5), (**E**) serum GLP-1 concentrations after 8 weeks (mean ± SEM; *n* = 3). Values with different letters are significantly different: *p* < 0.05 (a > b > c).

**Figure 3 biomedicines-13-02077-f003:**
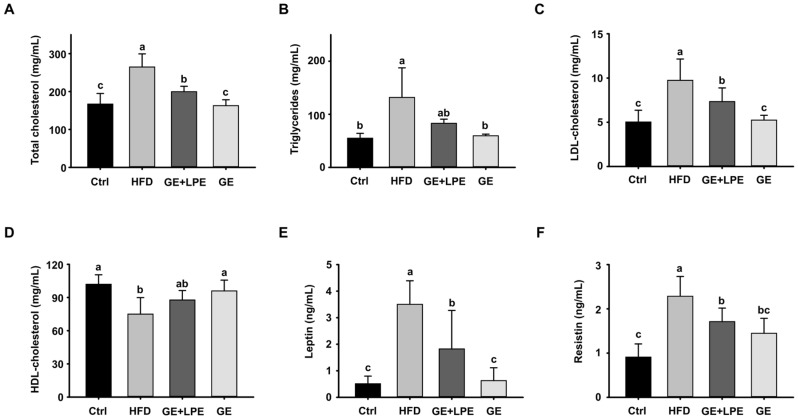
Effects of GE and GE+LPE on lipid profile and adipokine concentrations of HFD-fed mice. (**A**) Total cholesterol (mean ± SEM; *n* = 10), (**B**) Triglyceride (mean ± SEM; *n* = 6), (**C**) LDL-cholesterol (mean ± SEM; *n* = 12), (**D**) HDL-cholesterol (mean ± SEM; *n* = 8), (**E**) Leptin (mean ± SEM; *n* = 7), and (**F**) Resistin (mean ± SEM; *n* = 5) concentrations after 8-week treatment. Values with different letters are significantly different: *p* < 0.05 (a > b > c).

**Figure 4 biomedicines-13-02077-f004:**
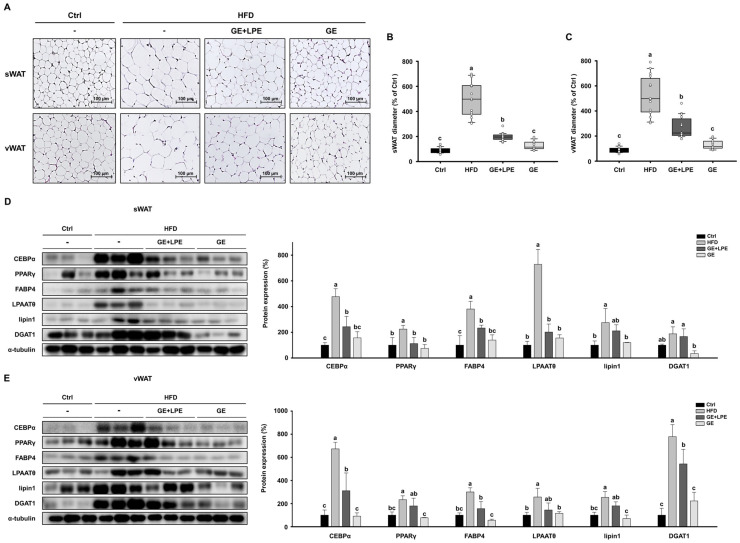
Effects of GE and GE+LPE on adipocyte size and the expression of adipogenic/lipogenic proteins in WAT of HFD-fed mice. (**A**) Sections of sWAT and vWAT stained with hematoxylin and eosin. Quantification of adipocyte size (**B**) in the sWAT (mean ± SEM; *n* = 14) and (**C**) in the vWAT (mean ± SEM; *n* = 14). (**D**) Western blots of adipogenic proteins (C/EBPα, PPARγ, and FABP4) and lipogenic proteins (LPAATθ, lipin1, and DGAT1) in sWAT. (**E**) Western blots of adipogenic proteins (C/EBPα, PPARγ, and FABP4) and lipogenic proteins (LPAATθ, lipin1, and DGAT1) in vWAT. Protein expression levels were normalized to α-tubulin. Values with different letters are significantly different: *p* < 0.05 (a > b > c).

**Figure 5 biomedicines-13-02077-f005:**
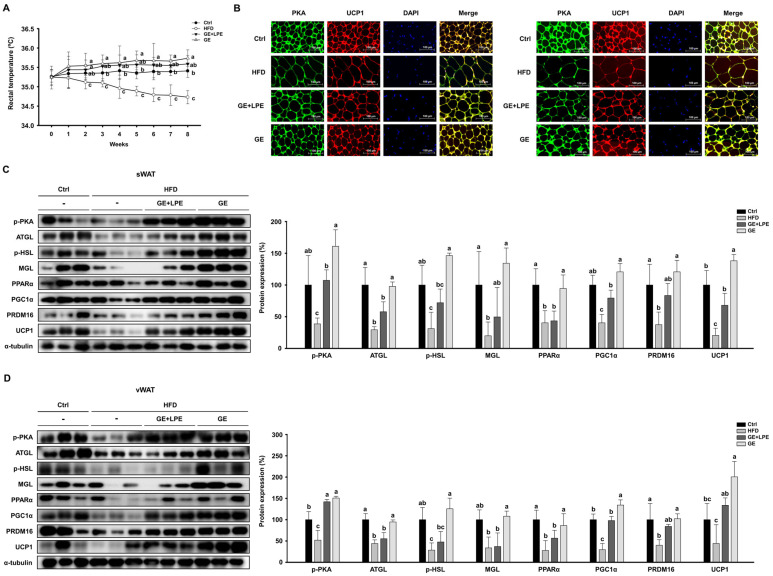
Effects of GE and GE+LPE on lipolysis and browning in the WAT of HFD-fed mice. (**A**) Rectal temperatures of the mice during 8 weeks of treatment (mean ± SEM; *n* = 14) (**B**) Immunofluorescence staining of PKA and UCP1 in sWAT and vWAT. (**C**) Western blots of proteins involved in lipolysis (p-PKA, ATGL, p-HSL, and MGL) and browning (PPARα, PGC1α, PRDM16, and UCP1) in sWAT. (**D**) Western blots of proteins involved in lipolysis (p-PKA, ATGL, p-HSL, and MGL) and browning (PPARα, PGC1α, PRDM16, and UCP1) in vWAT. Protein expression levels were normalized to α-tubulin. Data are expressed as mean ± SEM. Values with different letters are significantly different: *p* < 0.05 (a > b > c).

**Figure 6 biomedicines-13-02077-f006:**
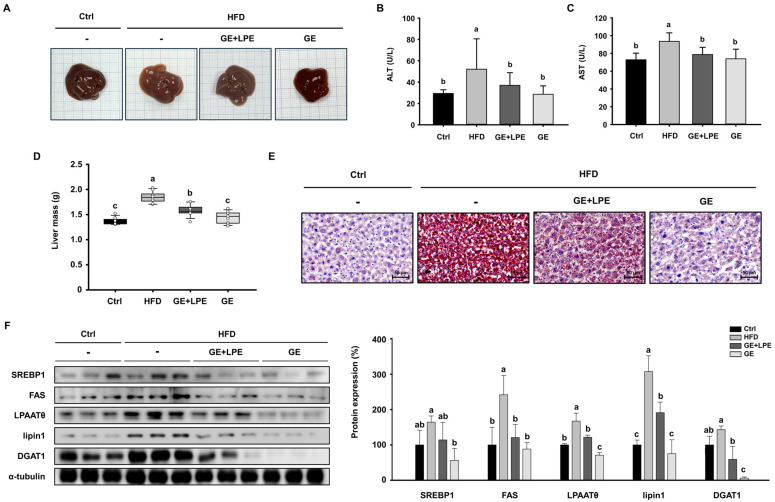
Effects of GE and GE+LPE on hepatic lipid accumulation in HFD-fed mice. (**A**) Representative images of mouse liver morphology. (**B**) Serum ALT (mean ± SEM; *n* = 10) and (**C**) AST concentrations (mean ± SEM; *n* = 7). (**D**) Liver mass (mean ± SEM; *n* = 8). (**E**) Oil red O-stained liver sections. (**F**) Western blots of hepatic lipogenic protein expression (SREBP1, FAS, LPAATθ, lipin1, and DGAT1). Protein expression levels were normalized to α-tubulin. Data are expressed as mean ± SEM. Values with different letters are significantly different: *p* < 0.05 (a > b > c).

**Table 1 biomedicines-13-02077-t001:** Composition of Ginger Extract (GE) and Long Pepper Extract (LPE).

Component	Ginger Extract (GE)	Long Pepper Extract (LPE)
Energy (kcal/100 g)	623.49	357.85
Carbohydrate (%)	39.96	86.30
Crude fat (%)	51.21	0.45
Crude protein (%)	0.69	2.15
Ash (%)	5.10	5.38
Moisture (%)	3.04	5.72
Total polyphenols (GAE mg/g)	139.4 ± 2.7	5.6 ± 0.1
Marker compound content (mg/g)	6-gingerol: 132	piperine: 20

## Data Availability

All data generated or analyzed in this study are included in the article and Appendix A. Further inquiries can be directed to the corresponding author.

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
