# Peer review of "Comparison of Anti-Obesity Effects of Ginger Extract Alone and Mixed with Long Pepper Extract"

_biomedicines, 2025, doi:10.3390/biomedicines13092077_

Round 1

Reviewer 1 Report

Comments and Suggestions for Authors

i. In the methodology section, authors should mention the  breed of the mice used in the study.

ii. Section 2.2 of the methodology section should be completely reworked to reflect the days of treatment, days of feeding mice with HFD before treatment began.

iii. Authors should justify the reason for measuring the blood glucose level of the obese mice at the time interval stated. Ideally, the blood glucose level should be measured at least once in a week.

iv. In the discussion section, authors should compare the results of the extract used with those reported in the literature. Several studies has been reported on Ginger along this research line. Authors should improve their literature survey and include relevant references in the revised manuscript.

v. The conclusion does not reflect the unique findings of the study. Authors should present key findings and recommendations for future studies.

Comments on the Quality of English Language

The quality of the English language should be improved

Reviewer 2 Report

Comments and Suggestions for Authors
  1. Section 2.1 does not provide detailed specifications for the extraction parameters used in supercritical COâ‚‚ processing, such as pressure, temperature, and duration.
  2. It is suggested that Figure 1 be presented as an attachment.
  3. A critical issue is that the WB of PPAR in Figure 5D is inconsistent with theoriginal images for blots provided in the attachment. Please confirm. Please also confirm each of the other original images for blots.
  4. Figure 3D indicates a sample size of n = 5 for serum insulin, and Figure 4F presents a sample size of n = 5 for resistin. However, the methodology section specifies that each group comprises n = 14. Furthermore, the sample sizes across Figures 2, 3, and 4 are inconsistent. Please provide an explanation for the discrepancy in sample size.
  5. The main issue addressed here is the overinterpretation of the experimental results. While the paper asserts that the ginger extract is "significantly superior" to the mixture, the experimental design does not provide sufficient evidence to support such a definitive conclusion.
  6. This manuscript demonstrates a degree of innovation in its research design. However, substantial revisions are required in the areas of methodological detail, data rigor, and the presentation of conclusions to align with publication standards.

Reviewer 3 Report

Comments and Suggestions for Authors

Comparison of Anti-Obesity Effects of Ginger Extract Alone and Mixed with Long Pepper Extract              

Obesity is pandemic. Theme is welcome. Study design is in accordance with the ethical principles but the study group should be larger. Introduction could be improved. Methods include Fasting blood glucose measurement (OGTT), Immunofluorescence staining, biochemical analysis as well as statistics. Figure 1 represents the HPLC chromatogram. Figures 2-6 reveal the effects of ginger extract (GE) and GE+ long pepper extract (LPE) on body weight. Conclusion are supported by the results which are clearly presented. References must be up dated.

Round 2

Reviewer 1 Report

Comments and Suggestions for Authors

No comment 

Reviewer 2 Report

Comments and Suggestions for Authors

The authors have made detailed revisions and explanations for the problems present in the manuscript, and have recommended that the journal accept this manuscript.

Reviewer 3 Report

Comments and Suggestions for Authors

Article could be published